# Risky Driver Recognition with Class Imbalance Data and Automated Machine Learning Framework

**DOI:** 10.3390/ijerph18147534

**Published:** 2021-07-15

**Authors:** Ke Wang, Qingwen Xue, Jian John Lu

**Affiliations:** Key Laboratory of Road and Traffic Engineering of the State Ministry of Education, College of Transportation Engineering, Tongji University, Shanghai 201804, China; kew@tongji.edu.cn (K.W.); 1710517@tongji.edu.cn (Q.X.)

**Keywords:** risky driving, automated machine learning, imbalanced data, sampling, cost-sensitive learning, probability calibration

## Abstract

Identifying high-risk drivers before an accident happens is necessary for traffic accident control and prevention. Due to the class-imbalance nature of driving data, high-risk samples as the minority class are usually ill-treated by standard classification algorithms. Instead of applying preset sampling or cost-sensitive learning, this paper proposes a novel automated machine learning framework that simultaneously and automatically searches for the optimal sampling, cost-sensitive loss function, and probability calibration to handle class-imbalance problem in recognition of risky drivers. The hyperparameters that control sampling ratio and class weight, along with other hyperparameters, are optimized by Bayesian optimization. To demonstrate the performance of the proposed automated learning framework, we establish a risky driver recognition model as a case study, using video-extracted vehicle trajectory data of 2427 private cars on a German highway. Based on rear-end collision risk evaluation, only 4.29% of all drivers are labeled as risky drivers. The inputs of the recognition model are the discrete Fourier transform coefficients of target vehicle’s longitudinal speed, lateral speed, and the gap between the target vehicle and its preceding vehicle. Among 12 sampling methods, 2 cost-sensitive loss functions, and 2 probability calibration methods, the result of automated machine learning is consistent with manual searching but much more computation-efficient. We find that the combination of Support Vector Machine-based Synthetic Minority Oversampling TEchnique (SVMSMOTE) sampling, cost-sensitive cross-entropy loss function, and isotonic regression can significantly improve the recognition ability and reduce the error of predicted probability.

## 1. Introduction

According to the statistics of historical accidents in road traffic, risky driving behavior is the leading cause of traffic insecurity [1]. Risky driving behavior refers to a series of irregular traffic behaviors and violations of traffic rules to realize the driver’s driving intention in driving on the road. The quantification and identification of risky driving behaviors and risky drivers are crucial for road traffic safety.

Most research on risky driving and risky driver recognition algorithms focuses on risky driving state recognition, including aggressive driving, distracted driving, fatigue driving, etc. For example, Wang et al. [2] used discrete Fourier coefficients of vehicle trajectory data, such as distance between vehicles and speed, as input and used imbalanced class boosting algorithms to identify aggressive car-following drivers. Sun et al. [3] combined background features such as driving time and sleep time to establish a dual-layer fusion fatigue driving recognition model based on the driver’s facial features and operating characteristics. Liu et al. [4] conducted a natural driving experiment and used the driver’s eye movement and hand movement data to establish a semi-supervised learning model for distracted driving. Chandrasiri et al. [5] used a driving simulator to observe the driver’s vehicle manipulation data on various turning radius roads and established a classification model of driving skill level [5]. The primary data used by the risky driving recognition model include driver facial data [3], hand data [6], posture data [7], physiological data [8], steering wheel angle [5], pedal data [9], vehicle driving trajectory data [10], etc. Naturalistic driving data collection mainly relies on experiment vehicles with equipment collecting vehicle operation data from Controller Area Network (CAN) [11] or onboard motion sensors [12]. Some studies also use a driving simulator to observe the driving behavior of experimenters in a pre-designed driving environment [13,14]. In addition, cameras deployed on the roadside or unmanned aerial vehicles (UAV) can record traffic video, from which advanced computer vision algorithms can extract vehicle trajectory data at a lower cost compared to naturalistic driving experiment and driving simulator [2,10]. A smartphone is equipped with multiple sensors, including an accelerometer, gyroscope, magnetometer, microphone, cameras, thermometer, and Global Positioning System (GPS). Thus, the smartphone has the ability to track a vehicle’s motion and location and has been applied to detect abnormal driving behaviors [15,16].

There is no such thing as a free lunch. Although vehicle trajectory data can be easily extracted from traffic video, the challenge of using this type of data in a risky driving recognition study is data labeling. The data labeling of naturalistic driving data and driving simulator experiment data is straightforward. Some studies distinguish between normal driving samples and risky driving samples by observing illegal driving behavior [17] or accident [18] in naturalistic driving data. Subjective data labeling methods include experts scoring [19] and quantitative questionnaires on risky driving behavior [20]. None of these methods are suitable for a large sample size video-extracted vehicle trajectory. Xue [10] met the challenge by using collision surrogate measurements such as Time to Collision (TTC) and Margin to Collision (MTC) to distinguish between drivers in risky car-following states and drivers in normal car-following states. Besides, clustering algorithms [21] or semi-supervised learning [4,22] do not require or only partially require data labeling, but the results often lack reliable verification standards.

In driving behavior research, data is often imbalanced. For example, the incidence of risky driving behaviors is much less than normal driving behaviors in real traffic. Supervised learning algorithms pay more attention to normal driving behavior, which is the majority, and have a poor predictive performance for the minority class. It is a problem since usually recognizing the minority class is the goal. Existing studies generally pre-sample imbalanced data to reduce imbalance before using supervised learning algorithms [23]. There are also studies using cost-sensitive learning to increase the classification error cost for the minority class to compensate for the bias in the class distribution [24]. Ensemble learning algorithms that combine sampling with bagging or boosting, such as Synthetic Minority Oversampling TEchnique Boosting (SMOTEBoost), Random UnderSampling Boosting (RUSBoost), and EasyEnsemble, perform better than traditional ensemble learning without sampling [25]. Cost-sensitive learning, including example weighting and threshold moving, was compared to sampling methods in some studies, and there is no clear winner [26,27]. Every method has strengths and weaknesses. There is no universal primary choice since imbalanced data structure varies, let alone state-of-art classifiers like eXtreme Gradient Boosting (XGBoost) may differ from traditional ones, such as decision trees, K-Nearest Neighbors (KNN), and Support Vector Machine (SVM).

Some studies combine sampling and cost-sensitivity to solve the imbalance problem, but none are in the field of risky driving recognition. For example, Le et al. [28] combined sampling with Cluster-based Boosting (CBoost) algorithm, which is based on the cost-sensitive learning framework, to predict bankruptcy. Peng et al. [29] applied sampling and cost-sensitive MLP to predict traffic accidents. The limitation of these studies is preset of sampling method, sampling ratio, and example weights in cost-sensitive classifiers. Applying a fixed combination of sampling method, sampling ratio, and example weights, the machine learning algorithm may end at suboptimal solutions and perform even worse than using sampling or cost-sensitive learning alone. 

Automated machine learning (AutoML) refers to the automated process of machine learning model development, including but not limited to data cleaning and processing, feature extraction and selection, model selection, and parameter selection. AutoML reduces the human effort necessary for applying machine learning. Given a dataset, AutoML automatically and simultaneously chooses algorithms and sets their hyperparameters to optimize empirical performance. This paper aims to build an AutoML framework that can automatically select the best sampling method, cost-sensitive loss function, probability calibration method, and the corresponding hyperparameters to establish a risky driver recognition model. A UAV video-extracted vehicle trajectory dataset collected from a German highway is used to train the model. The performance of 12 sampling methods, 2 cost-sensitive loss functions, and 2 probability calibration methods are included in the automated machine learning, and their performance is compared. 

This paper’s remainder is organized as follows: Section 2 describes the vehicle trajectory data used for modeling; Section 3 introduces the framework of the risky driver recognition modeling process and the methodology of each part in the framework; Section 4 presents and discusses the results; Section 5 concludes.

## 2. Data

To demonstrate the proposed automated machine learning framework, we use the vehicle trajectory data from the Highway Drone Dataset (highD) [30] to establish a risky driver recognition model as a case study. Traffic was recorded at six German highways using UAV, from which 110,500 vehicle trajectories were extracted using state-of-the-art Computer Vision algorithms. The UAV camera can cover a 420-m length of highway with a typical vehicle positioning error of less than 10 cm. The recorded traffic video has 25 frames per second. The vehicle’s position was detected and tracked every 0.04 s and smoothed using Bayesian smoothing. Other driving information, including speed, acceleration, lane-changing, and car-following, can be derived from vehicle position. The data used in this paper were recorded on a 6-lane highway during the morning traffic peak period. The trajectory of 2850 vehicles was recorded over 19 min 38 s. Among all vehicles, 2427 private cars with recorded trajectories longer than 10 s were kept in the following studies.

## 3. Methodology

The research framework of this paper is present in Figure 1. First, vehicle trajectory is analyzed to label each driver as a risky driver or normal driver based on its collision risk. The collision risk evaluation method is introduced in Section 3.1. Once risky drivers are labeled based on ACR, we try to establish a risky driver recognition model using less trajectory information. Section 3.2 describes how to extract and select features from the trajectory. Since the number of risky drivers and the number of normal drivers is imbalanced, we apply three class-imbalance techniques: sampling, cost-sensitive learning, and probability calibration. These three techniques are introduced in Section 3.3, Section 3.4 and Section 3.5, respectively. Section 3.6 covers automated machine learning. Section 3.7 explains how we evaluate model performance.

### 3.1. Collision Risk Evaluation

Rear-end crashes are the most frequently occurring type of collision and account for approximately 29% of all crashes, according to U.S. Department of Transportation traffic crash statistics [31]. Based on the target vehicle’s moving state, we calculate the collision risk following the rules below:Car-following: if the target vehicle has a leading vehicle within 50 m, calculate the rear-end collision risk between the target vehicle and the leading vehicle (shown in Figure 2a).Lane-changing: if the target vehicle’s center crosses a lane line (shown in Figure 2b), we start calculating the rear-end collision between the target vehicle and its leading vehicle and between the target vehicle and its following vehicle until the target vehicle’s land-changing is completed, when the target vehicle’s distance to the lane line is greater than 0.5 m (shown in Figure 2c).

For each vehicle, rear-end collision at time t is calculated based on its Difference of Space distance and Stopping distance (*DSS*): (1)$$DSS(t)=\frac{v_{l}^{2}(t)-v_{f}^{2}(t)}{2\mu g}+d(t)-\tau v_{f}(t)\;$$
where *v_l_* and *v_f_* are the longitudinal speed of the leading and following vehicles, respectively; *μ* is the fraction rate, set to 0.7; *g* is the acceleration of gravity, 9.8 m/s^2^; *d* is the longitudinal gap between the leading and following vehicles; *τ* is the reaction time of drivers. When the following vehicle is accelerating, *τ* is set to 1.5 s. When the following vehicle is decelerating or idling, *τ* is set to 0.7 s.

When *DSS* ≥ 0, it means the following vehicle has enough time to decelerate and avoid a collision. When *DSS* < 0, the following vehicle has a collision risk. Wang [2] proposed measurement of Collision Risk (*CR*) as the absolute value of *DSS* divided by the following vehicle’s speed.
(2)$$CR(t)=\{\begin{array}{cc} 0 & \mathrm{if}\;DSS(t)\;\geq \;0\\ \left|DSS(t)\right|/v_{f}(t) & \mathrm{if}\;DSS(t)\;<\;0 \end{array}\;$$

When *CR*(*t*) > 0, the following vehicle does not have enough time to react to the leading vehicle’s abrupt deceleration at time *t*. *CR*(*t*) reflects the extra time the following vehicle needs to avoid a collision. To measure the overall collision risk exposed to the target vehicle over the whole trajectory, we calculate the average collision risk (ACR) as follows:(3)$$\mathrm{ACR}=\frac{1}{T}\sum_{t=0}^{T}\left[CR_{L}(t)+CR_{F}(t)\right]\Delta t\;$$
where *T* is the observation duration of the target vehicle; Δ*t* is the sampling interval, 0.04 s; *CR*_L_(*t*) is the rear-end collision risk between the target vehicle and its leading vehicle; *CR*_F_(*t*) is the rear-end collision risk between the target vehicle and its following vehicle in the target lane if the target vehicle is in the lane-changing process; otherwise, *CR*_F_(*t*) = 0.

ACR is a metric of collision risk for individual vehicles. Certainly, drivers with ACR = 0 are safe since they have no collision risk during observation period. How to label drivers with a positive ACR value? Drivers with ACR larger than a threshold are labeled as risky drivers. We determine the threshold using the Interquartile Range (IQR) method, which was proposed by Laurikkala et al. [32]. It is a common method in outlier detection and can be used to calculate the threshold of abnormal data under various distribution [33]. The threshold can be calculated as follows.
(4)$$\mathrm{Threshold}=Q_{3}\;+\;1.5\left(Q_{3}-Q_{1}\right)$$
where *Q*_3_ is the upper quartile of the non-zero ACR distribution; *Q*_1_ is the lower quartile of the non-zero ACR distribution.

### 3.2. Feature Extraction and Selection

Longitudinal speed, lateral speed, and gap are chosen to recognize risky drivers. Discrete Fourier Transform (DFT) or Fast Fourier Transform (FFT) has been applied in many driving behavior studies [16,34,35,36] to convert the time series of driving features to signal amplitude in the frequency domain. Xue et al. [10] found that DFT is a better feature extraction method than statistical parameters, such as mean, standard deviation, maximum, and minimum. 

The DFT of a given time series (*x_1_*, *x_2_*, …, *x_N_*) is defined as a sequence of N complex numbers $\left({\mathrm{DFT}}_{0},{\;\mathrm{DFT}}_{1},\ldots ,{\;\mathrm{DFT}}_{N-1}\right)$:(5)$${\mathrm{DFT}}_{k}=\sum_{n=0}^{N-1}x_{n}e^{\left(-\frac{2\pi i}{N}kn\right)}$$
where *i* is the imaginary unit.

For longitudinal speed, lateral speed, and gap, each time series is converted to 20 DFT coefficients. The mean, standard deviation, and coefficient of variation of longitudinal speed, lateral speed, and gap are also included as features. Therefore, each driver has 72 features in total. Figure 3 shows two numerical examples of longitudinal speed and processed DFT coefficients. Figure 3a,c shows the longitudinal speed data of two vehicles (ID: 25-300 and 25-591) in the time domain, and Figure 3b,d shows the corresponding DFT coefficients. The longitudinal speed of vehicle 25-300 is more volatile than that of vehicle 25-591. Therefore, vehicle 25-300 should have higher amplitudes in high frequency than vehicle 25-591, while vehicle 25-591 has higher amplitudes in low frequency than vehicle 25-300. In the frequency domain, we can observe that vehicle 25-300 has lower amplitude in frequency between 0.04 and 0.08 Hz and higher amplitude in frequency between 0.12 and 0.32 Hz, compared to vehicle 25-591. Figure 3 shows that DFT can reveal signal amplitudes at each frequency hidden in time series data regardless of the length of time series.

Recursive Feature Elimination (RFE) is a feature selection algorithm. First, a full risky driver recognition model using all 72 features is created. Second, features are ranked from most important to least based on their feature importance. Once at a time, the least important feature is iteratively eliminated prior to retraining the model. The iteration continues until the model’s performance cannot improve or no features are left in the pool.

### 3.3. Sampling Methods

We consider five undersampling methods: Random UnderSampling (RUS), Tomek links, Edited Nearest-Neighbors (ENN), Repeated Edited Nearest-Neighbors (RENN), and All-K-Nearest-Neighbors (AllKNN). RUS randomly removes instances from the majority class until the imbalance class ratio reaches the desired level. Tomek links are pairs of very close instances but of opposite classes. Removing the instances of the majority class of each pair increases the space between the two classes, facilitating the classification process. ENN requires a sample (usually from the majority class) to have an opposite class instance as its nearest neighbor in order to remove it. By contrast, Tomek Links requires both samples to be each other’s nearest neighbors. In summary, Tomek Links uses a more restrictive condition resulting in fewer samples being removed. RENN repeats ENN several times. AllKNN increases the k value in k-nearest neighbors while repeating ENN. Except for RUS, all the other under-sampling methods cannot control the number of samples in the minority class over the number of samples in the majority class after resampling. Generally, the data is still imbalanced after sampling by Tomek links, ENN, RENN, or AllKNN.

We consider five oversampling methods: Random Over Sampling (ROS), Synthetic Minority Oversampling Technique (SMOTE), SVMSMOTE, Borderline-SMOTE, and Adaptive Synthetic Sampling (ADASYN). ROS randomly replicates the minority class examples until the imbalance class ratio reaches the desired level. SMOTE generates a new synthetic example by linear interpolation between a randomly selected sample and one of its neighbors in the feature space. There are variants of SMOTE sampling which have different rules to choose samples and neighbors. SVMSMOTE uses an SVM algorithm to establish a boundary between classes and generate new synthetic samples near borderlines. Borderline-SMOTE uses K-nearest neighbors instead of SVM, to identify the misclassified samples around the decision boundary. Adaptive Synthetic Sampling (ADASYN) generates more samples around “harder-to-learn” minority samples that have more majority neighbors. All these oversampling methods have the option to set a sampling rate and reach the desired class ratio.

Oversampling and undersampling have their drawbacks. Oversampling may introduce noises to the data. Undersampling removes useful information from the majority class. The hybrid sampling method combines oversampling with undersampling to overcome their drawbacks. SMOTEENN generates new synthetic examples first using SMOTE and then conduct undersampling using ENN. The difference between SMOTETomek and SMOTEENN is that SMOTETomek uses Tomek Links as the undersampling method instead of ENN.

We propose a comprehensive automated machine learning framework that combines cost-sensitive learning and sampling. The sampling method partially balances data before classifier training, and cost-sensitive XGBoost is trained with minority examples’ weight increased. With hyperparameter optimization, we can optimize the sampling rate and example weights and create a more flexible way to handle imbalanced data.

### 3.4. Cost-Sensitive XGBoost Loss Functions

XGBoost, which stands for eXtreme Gradient Boosting, is a Gradient Tree Boosting-based algorithm [37] that is superior in performance, fast in training time, and has an easy-to-use interface. XGBoost uses the cross-entropy loss (log loss) function, which is a probability-based metric, to measure the performance of classification. A generalized cross-entropy loss function for binary classification problem with example weight, weighted binary cross-entropy loss, can be expressed as:(6)$$L_{w}=-\sum_{i=1}^{N}\left[wy_{i}\log\left({\hat{p}}_{i}\right)+\left(1-y_{i}\right)\log\left(1-{\hat{p}}_{i}\right)\right]$$
where *y_i_* is the true class of sample *i*, *y_i_* = 1 for positive (minority) instances, and 0 for negative (majority) instances. *w* is the weight of the minority class. If w is greater than 1, an extra loss is added on positive (minority) instances. *N* is the sample size. ${\hat{p}}_{i}$ is the estimated probability of being positive for sample *i*.

Focal loss function was proposed by Lin [38] to solve the imbalanced foreground-background class problem encountered in dense objective detection. The focal loss gives more weight to “hard examples” whose estimated probability is far away from the true class (for binary classification, ${\hat{p}}_{i}$ is close to 0 when *y_i_* = 1, and ${\hat{p}}_{i}$ is close to 1 when *y_i_* = 0). In the case of class imbalance, the number of positive instances is much less than the number of negative instances. Since the machine learning algorithms may overcompensate and give too much focus to the negative (majority) class, the “hard examples” are mainly positive instances misclassified as negative instances.

Wang [39] proposed Imbalance-XGBoost, which combines XGBoost with the weighted focal loss for the class-imbalanced problem. Weighted binary focal loss can be denoted as:(7)$$L_{f}=-\sum_{i=1}^{N}\left[wy_{i}{\left(1-{\hat{p}}_{i}\right)}^{\gamma }\log\left({\hat{p}}_{i}\right)+{\left(1-y_{i}\right){\hat{p}}_{i}}^{\gamma }\log\left(1-{\hat{p}}_{i}\right)\right]$$
where *γ* is the focusing parameter, *γ* ≥ 0. If *γ* = 0, then Equation (7) will be the same as Equation (6).

### 3.5. Probability Calibration

Many machine learning algorithms not only predict class but also can predict a probability or a probability-like score for each class. The predicted probability as a measure of uncertainty can be used to evaluate models when only predicting class is not sufficient to calculate Receiver Operating Characteristic (ROC) curve and Precision-Recall curve.

There are two main reasons that probability calibration is needed for imbalanced data. First, algorithms like SVM, boosted trees are not trained using a probabilistic framework and do not provide calibrated probabilities [40]. Second, supervised learning trained with imbalanced data systematically underestimates the probabilities for minority class instances [41].

There are two common probability calibration methods: Platt scaling and isotonic regression. Platt [42] introduced the calibration method Platt scaling, which can train a logistic regression to map the original classifier’s output to the true class probability. Isotonic regression is a non-parametric approach introduced by Zadrozny and Elkan [43,44]. Isotonic regression fits a piecewise constant non-decreasing function, where predicted probabilities or scores in each bin are assigned the same calibrated probability that is monotonically increasing over bins.

Platt scaling is preferable when the distortion in the predicted probabilities is sigmoid-shaped. Isotonic regression is a more powerful calibration method that can correct any monotonic distortion. However, isotonic regression may perform worse than Platt scaling when calibration data has a small sample size.

### 3.6. Automated Machine Learning

We propose an AutoML framework that automatically and simultaneously selects the sampling method, sampling ratio, cost-sensitive loss function, minority class weight, and probability calibration method to maximize the evaluation metrics of risky driving recognition model. These five elements are shown in Figure 4, with the candidates available in each element. Since the imbalance ratio of the dataset used in this paper is 1:22.3, the sampling ratio is set to be from 1 to 22.3. Sampling_ratio does not apply to TomekLinks, ENN, RENN, and AllKNN since these sampling methods cannot control the sampling rate. Sampling ratio is determined by dividing the number of majority examples by the number of minority examples after sampling. When sampling ratio is 1, the dataset is exactly balanced; when sampling ratio is 22.3, the dataset is not sampled since its imbalance ratio is unchanged. Therefore, by choosing the optimal sampling ratio, the AutoML can control the degree of imbalance. If the optimal sampling ratio is 22.3, it indicates that it is better not to alleviate data imbalance by sampling. The higher the minority class weight is, the more important the minority examples are. The majority class weight is always 1. If the AutoML suggests the minority class weight equals 1, the loss function in XGBoost is not cost-sensitive since the weights of majority class and minority class are the same.

Besides the five elements mentioned above, there are hyperparameters to be determined, shown in Table 1. Manual tuning and grid search are usually enough for traditional classifiers with a small number of hyperparameters. However, for supervised learning algorithms with numerous hyperparameters, automated machine learning is more powerful because of its speediness, stability, and accuracy. We consider six hyperparameters in this paper: five hyperparameters for XGBoost classifier and one hyperparameter for focal loss. XGBoost has dozens of configurable hyperparameters, and we only consider the most important ones. 

HyperOpt [45] is a software project that provides automated algorithm configuration of the Scikit-learn machine learning library. Using Bayesian optimization, HyperOpt allows for the automatic search of the optimal value of the five elements shown in Figure 4 and hyperparameters listed in Table 1.

### 3.7. Cross-Validation and Evaluation Metrics

We use stratified 5-fold cross-validation to evaluate the classification algorithm’s performance. Stratified 5-fold cross-validation divides the 2427 vehicles randomly into five equal-sized subsets. Each subset has the same imbalance class ratio as the total dataset. At each time, three subsets are used for sampling and then training, one subset is used for probability calibration, and the last subset is used to test the performance of the trained model. This process rotates through each subset, and the average AUPRC, F1 score, precision rate, and recall rate represent the performance of the algorithm. To find the optimal automated machine learning result, we iterate the optimization process 500 times. The sampling method, loss function, probability calibration, and hyperparameter values that reach the highest average AUPRC after 500 iterations are the final result. As the optimal sampling method, loss function, probability calibration, and hyperparameters are chosen based on test data, we use a different stratified 5-fold cross-validation with the optimal results in the final evaluation to avoid overfitting. As the optimal results determined by the Bayesian optimization could end up at a local optimum, the whole procedure described above is repeated five times to get four different sets of optimal results and five different final evaluations. The average of five final evaluations will be presented in Section 4.

The performance of the recognition model depends on its power to identify risky drivers. This paper uses five important performance indices: precision rate, recall rate, f1 score, Area under the Precision-Recall Curve (AUPRC), and Expected Calibration Error (*ECE*).

Precision rate is defined as follows:(8)$$\mathrm{Precision}=\frac{\mathrm{TP}}{\mathrm{TP}+\mathrm{FP}}$$
where TP is the number of risky drivers correctly identified; FP is the number of normal drivers wrongly identified as risky drivers.

Recall rate is defined as follows:(9)$$\mathrm{Recall}=\frac{\mathrm{TP}}{\mathrm{TP}+\mathrm{FN}}$$
where FN is the number of risky drivers wrongly identified as normal drivers.

The F1 score is the harmonic average of precision rate and recall rate. A high F1 score represents high values in both precision rate and recall rate.
(10)$$F1=2\frac{\mathrm{Precision}\times \mathrm{Recall}}{\mathrm{Precision}+\mathrm{Recall}}$$

The precision-recall curve is a plot of the precision rate and the recall rate for different probability thresholds. When there is a class-imbalance problem, it is more appropriate to use Area Under Precision-Recall Curve (AUPRC) instead of Area Under Receiver Operating Characteristic curve (AUROC) to measure the model’s performance because AUROC with an imbalanced dataset might be deceptive and lead to over-optimistic evaluation of the model [46].

Expected Calibration Error (*ECE*) [47] is used to measure the miscalibration degree to which a model’s predicted probability departs from the true value.
(11)$$ECE=\sum_{k=1}^{M}P\left(k\right)\cdot \left|o_{k}-e_{k}\right|$$
where *o_k_* is the true fraction of positive instances in bin *k*, *e_k_* is the mean of the post-calibrated probabilities for the instances in bin *k*, and *P*(*k*) is the fraction of all instances that fall into bin *k*. The lower the values of *ECE*, the better is the calibration of a model.

## 4. Results and Discussion

### 4.1. Collision Risk and Risky Drivers

Using the method introduced in Section 3.1, we calculate the Average Collison Risk for 2427 private cars and plot them in Figure 5. The average of all ACRs is 0.036. Most vehicles have a zero or close-to-zero ACR value indicating they are driving at a safe state. Based on the IQR method, we find ACR = 0.5 as the risky driver threshold. We label drivers with ACR greater than 0.5 as risky drivers, accounting for 4.29% of all private car drivers. Therefore, the data is imbalanced, with 95.71% normal drivers and 4.29% risky drivers. The imbalance ratio is 1:22.3.

### 4.2. Automated Machine Learning Result

The risky driver recognition model was trained on a computer with an AMD Ryzen 1700X 8-core processor (3.40 GHz). A 500-iteration training takes about 30 min. After training, we found the best combination of sampling method, loss function, probability calibration method, and related hyperparameters. Two 5-fold cross-validations are involved. The first 5-fold cross-validation (denoted as CV1) was repetitively applied in the training process. We set the maximum number of automated learning iteration to 500, and the CV1 was repeated 500 times. The final learning results are determined based on their performance on CV1. The second 5-fold cross-validation (denoted as CV2) was used to generate the final evaluation results presented in this paper to avoid the overfitting problem. The difference between CV1 and CV2 is the random seed that impacts how samples are shuffled and split.

Figure 6 shows that AUPRC increases rapidly within the first 100 iterations and then slows down. As the searching continues, the AUPRC on test data in CV1 is still rising, even at a much slower pace. However, the AUPRC on test data in CV2 stays almost the same after 150 iterations. The gap between AUPRC of CV1 and CV2 is an indicator of the overfitting problem, and the gap grows after 1000 iterations. Therefore, setting the maximum number of searching iteration to 500 is enough for AutoML, and the model tends to overfit after 1000 iterations.

The AutoML results by HyperOpt depends on the initial values and may stop at a local optimum. We list the results of five independent rounds (each round contains 500 iterations of searching) of AutoML in Table 2. The optimal value of hyperparameters is not stable over each round. For example, the optimal number of estimators in XGBoost varies from 135 to 323; the weight of minority examples in loss function varies from 4.08 to 22.01. However, the variation of AUPRC in CV1 is small, ranging from 0.794 to 0.805. Unsurprisingly, AUPRCs in CV2 are lower than AUPRCs in CV1 and have a relatively wider but acceptable variation, ranging from 0.747 to 0.774. 

For all five rounds of AutoML, the best sampling method and loss function are SVMSMOTE and cross-entropy loss, respectively. There is no clear winner in probability calibration. In rounds 1 and 4, Platt scaling is the best probability calibration method, while in rounds 2 and 3, it is better to use the probability predicted by XGBoost directly without calibration. Isotonic regression is chosen by the AutoML in round 5. 

### 4.3. Manual Search of Class-Imbalance Handling Method

To prove the validity of AutoML results, we manually tested the performance of 13 different sampling methods (including no sampling), 2 loss functions, and 3 probability calibration methods (including no calibration) applied in the model training process. In total, there were 13 × 2 × 3 = 78 combinations, and we trained the risky driver recognition model 78 times independently, which took 1250 min. This section compares the performance of weighted (cost-sensitive) loss functions, and the sampling methods and probability calibration methods are analyzed in Section 4.4.

Figure 7a–c shows the AUPRC of risky driver recognition model with weighted focal loss and weighted cross-entropy loss. When no sampling is used, cross-entropy outperforms focal loss by having a higher AUPRC, regardless of the probability calibration method, shown in Figure 7a–c. However, there are certain combinations where focal loss is better than cross-entropy loss. For example, shown in Figure 7c, when using ADASYN, SMOTEENN, or ENN sampling and applying isotonic regression as the probability calibration method, AUPRC is improved when the cross-entropy loss function is switched to focal loss. 

The highest AUPRC of all 78 combinations is 0.763, generated by the combination of SVMSMOTE, cross-entropy, and no calibration. The second highest AUPRC is 0.758, generated by the combination of SVMSMOTE, cross-entropy, and Platt scaling. The result is consistent with the AutoML that SVMSMOTE is the best sampling method and cross-entropy loss is the best loss function for the risky driver recognition in this paper. The advantage of AutoML is its effectiveness and efficiency. AutoML can find the best class-imbalance handling method with 2.4% computational cost needed for manual search.

### 4.4. Comparison of Sampling Methods and Probability Calibrations

Setting cost-sensitive cross-entropy loss function, we plot AUPRC of different sampling methods and probability calibration methods in Figure 8. As shown in Figure 8, the best oversampling method is SVMSMOTE; the best undersampling method is Tomek Links; the best hybrid sampling method is SMOTETomek. Using cost-sensitive loss function alone without sampling is not a bad option, which beats several sampling + cost-sensitive combinations in terms of AUPRC. SMOTEENN + cost-sensitive has the worst AUPRC score, along with RENN and AllKNN.

Except for ROS, SVMSMOTE, and RUS, most sampling methods can get a higher AUPRC after probability calibration. For no sampling, Borderline-SMOTE, ADASYN, SMOTEENN, Tomek Links, and ENN, Platt scaling is the best calibration method in respect of AUPRC, compared to no calibration and isotonic regression. For SVMSMOTE, the AUPRC of isotonic regression and Platt scaling are lower but very close to the AUPRC of no calibration.

To find the best probability calibration method, we need help from another evaluation metric of probability calibration, Expected Calibration Error (*ECE*), which measures the error between predicted probabilities and empirical probabilities. As shown in Figure 9, Platt scaling and isotonic regression both reduce *ECE* substantially in most scenarios. The lowest *ECE* is reached by SVMSMOTE with isotonic regression. Therefore, we chose isotonic regression as the best probability calibration method.

### 4.5. Final Result

The weighted cross-entropy loss function and isotonic regression are shown to be the best loss function and probability calibration method, respectively. Results of different sampling methods with weighted cross-entropy loss function and isotonic regression are shown in Table 3. SVMSMOTE has the highest AUPRC, 0.758, and the lowest *ECE*, 0.015, among all sampling methods. The precision of SVMSMOTE is 0.797, and the recall rate is 0.536. Some other sampling methods have higher precision or recall rates, but since precision, recall, and F1 can be changed by threshold-moving, we chose the SVMSMOTE as the best sampling method based on AUPRC and *ECE* score. 

Many applications use undersampling or oversampling to create an exact-balanced dataset before model training. Exact-balanced sampling refers to sampling that generates data with equal-size minority and majority examples. Theoretically, sampling + cost-sensitive loss function is more flexible than using exact-balanced sampling alone. The hyperparameter “scale_pos_weight” in XGBoost controls the minority class weight in the cost-sensitive loss function. When “scale_pos_weight equals” 1, sampling + cost-sensitive loss function is equivalent to sampling only. When the sampling ratio equals 1, the data become exact-balanced after sampling. Figure 10 shows that sampling + cost-sensitive is better than exact-balanced sampling for oversampling, hybrid sampling, and RUS. As the minority sample size is minimal, exact-balanced RUS has a much worse AUPRC than other sampling methods. Tomek Links, ENN, RENN, and AllKNN are not considered in this section since they cannot create an exact-balanced dataset.

### 4.6. Discussion

In this paper, the ACR threshold value is determined using the IQR method. We tested the impact of different ACR threshold values on model’s evaluation metrics. As shown in Table 4, when using a smaller ACR threshold, the percentage of risky drivers increases and the evaluation metrics are improved in general, mainly because the data is less imbalanced. When the ACR threshold is 0.1, the imbalance ratio is 1:2.33, and the AUPRC is 0.934; when the ACR threshold is 0.6, the imbalance ratio is 1:64.57, and the AUPRC drops to 0.553.

The proposed automated machine learning framework is not limited to risky driver recognition but class imbalance problem in general. Other application domains suffer class imbalance problems, such as disease diagnosis, financial fraud detection, network intrusion detection, etc. Our future work will test the proposed framework’s performance on benchmark datasets from various fields.

## 5. Conclusions

This paper proposed an innovative AutoML framework that integrates sampling, cost-sensitive learning, and probability calibration with XGBoost to recognize risky drivers and combat the class imbalance problem. We found this combination more flexible and effective than using sampling or cost-sensitive learning alone to handle class imbalance problems. The AutoML framework can search for the best class-imbalance handling method out of 12 sampling methods, 2 cost-sensitive XGBoost loss functions, and 2 probability calibration methods. 

We used vehicle trajectory data to train the risky driver recognition model. Risky drivers were labeled based on their rear-end collision risk with surrounding vehicles. The inputs of the recognition model were the DFT coefficients of the target vehicle’s longitudinal speed, lateral speed, and the gap between the target vehicle and its preceding vehicle.

The optimal result learned by the AutoML framework was compared with the manual search result. Both agree that SVMSMOTE and weighted cross-entropy win the competition, but there is no clear answer of what probability calibration method is the best. In general, Platt scaling and isotonic regression can reduce the error between predicted probabilities and empirical probabilities. However, when combined with SVMSMOTE and weighted cross-entropy, the difference in AUPRC between probability calibration methods and no calibration is negligible. Finally, we chose isotonic regression as the probability calibration method used in model training since it has the lowest *ECE*.

Compared to manual searching, the AutoML can automatically find the optimal model pipeline and hyperparameters with significant savings on computational cost. A 500-iteration AutoML task in this paper takes only 30 min, which is 2.4% of the time needed for manual searching. We also found that the integration of sampling, cost-sensitive loss function, and probability calibration is more flexible and effective than using any class-imbalance handling method alone.

Future efforts should focus on the following aspects: (1) a more comprehensive collision risk evaluation on the target vehicle is needed to establish more reliable ground truth. (2) The framework proposed in this paper can be extended to other machine learning algorithms, such as deep neural networks, bagging and stacking of XGBoost classifiers.

## Figures and Tables

**Figure 1 ijerph-18-07534-f001:**
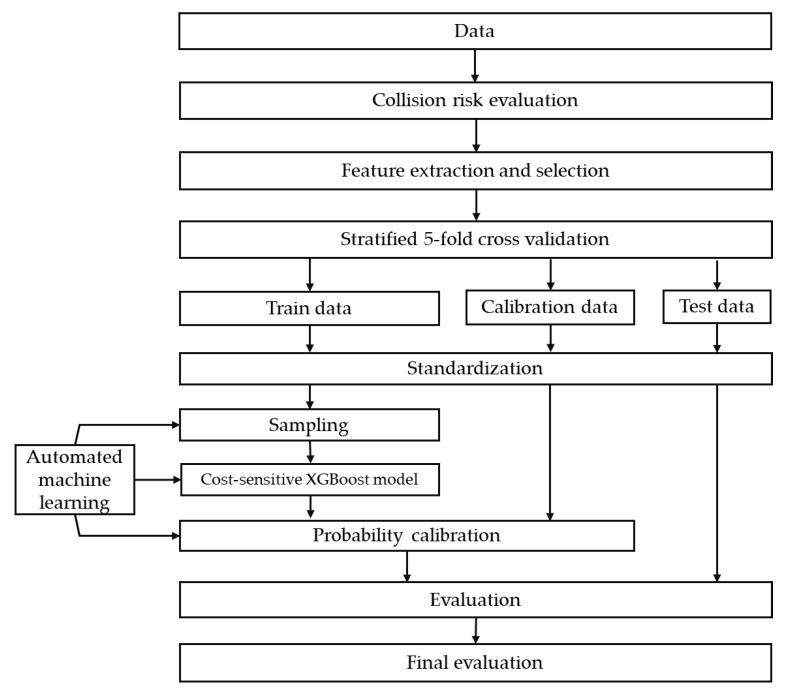
Methodology framework.

**Figure 2 ijerph-18-07534-f002:**
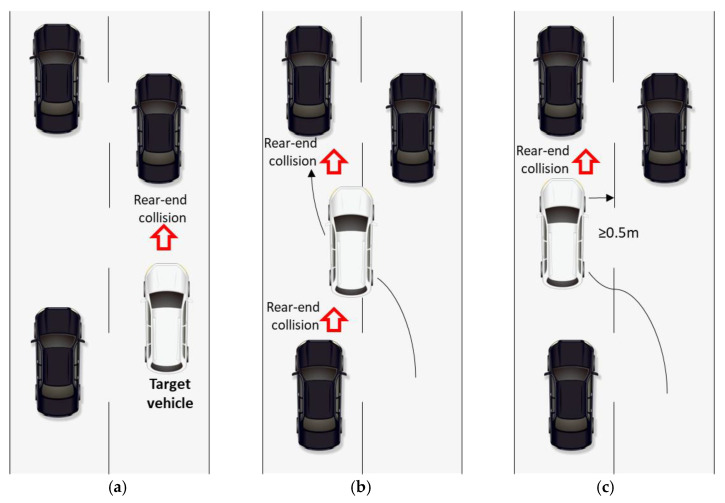
Rear-end collision evaluation for the target vehicle.

**Figure 3 ijerph-18-07534-f003:**
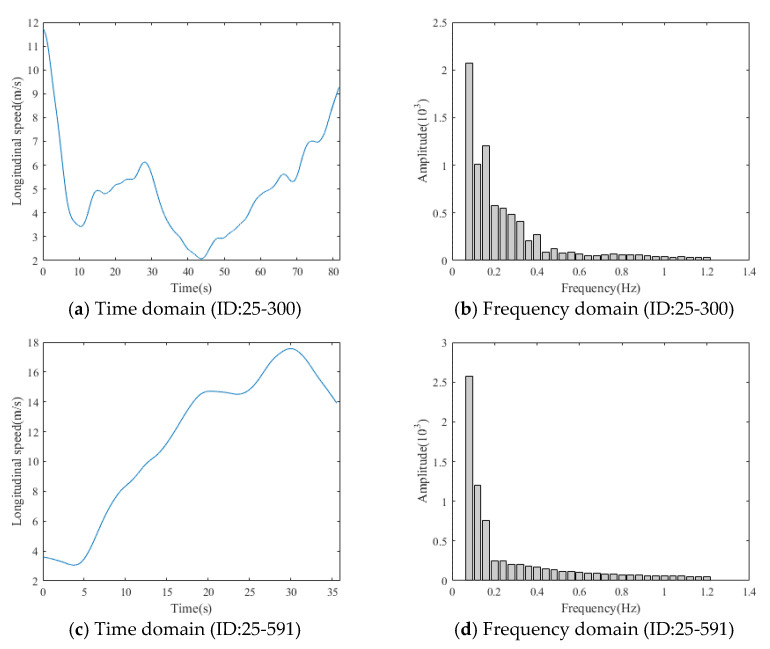
Rear-end collision evaluation for the target vehicle.

**Figure 4 ijerph-18-07534-f004:**
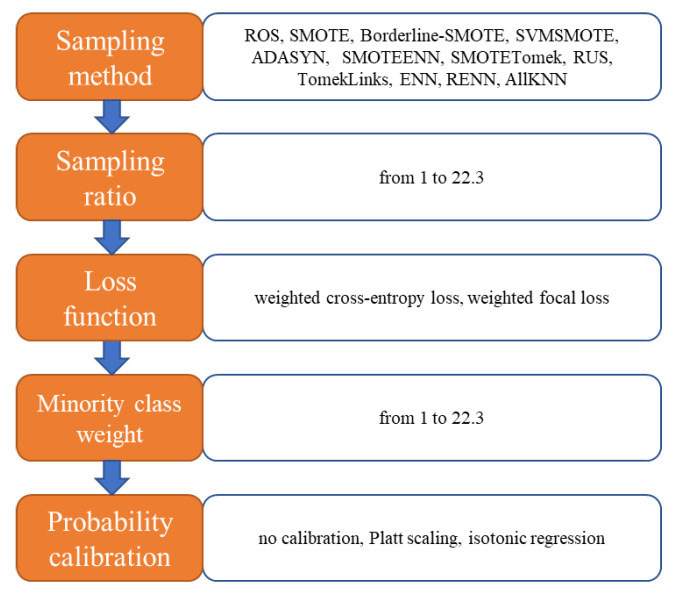
Five elements in the automated machine learning framework.

**Figure 5 ijerph-18-07534-f005:**
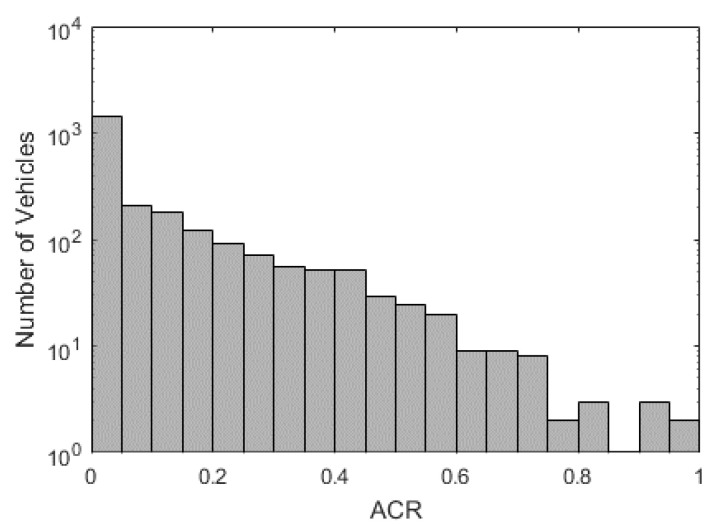
Histogram of all vehicles’ ACR.

**Figure 6 ijerph-18-07534-f006:**
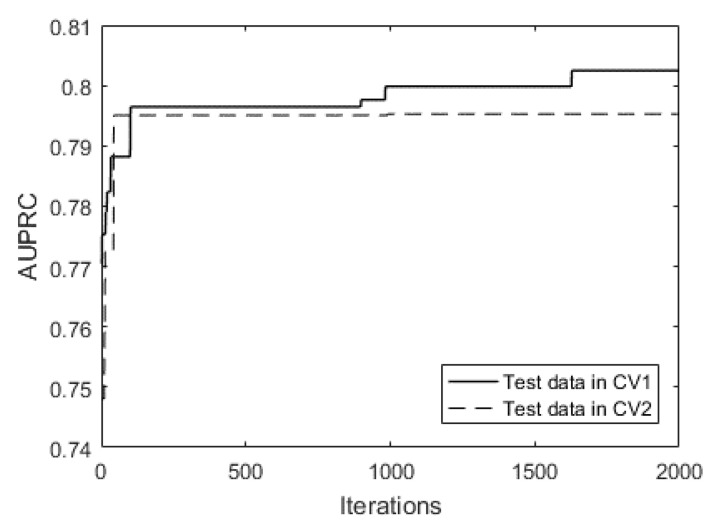
AUPRC of CV1 and CV2 over iterations.

**Figure 7 ijerph-18-07534-f007:**
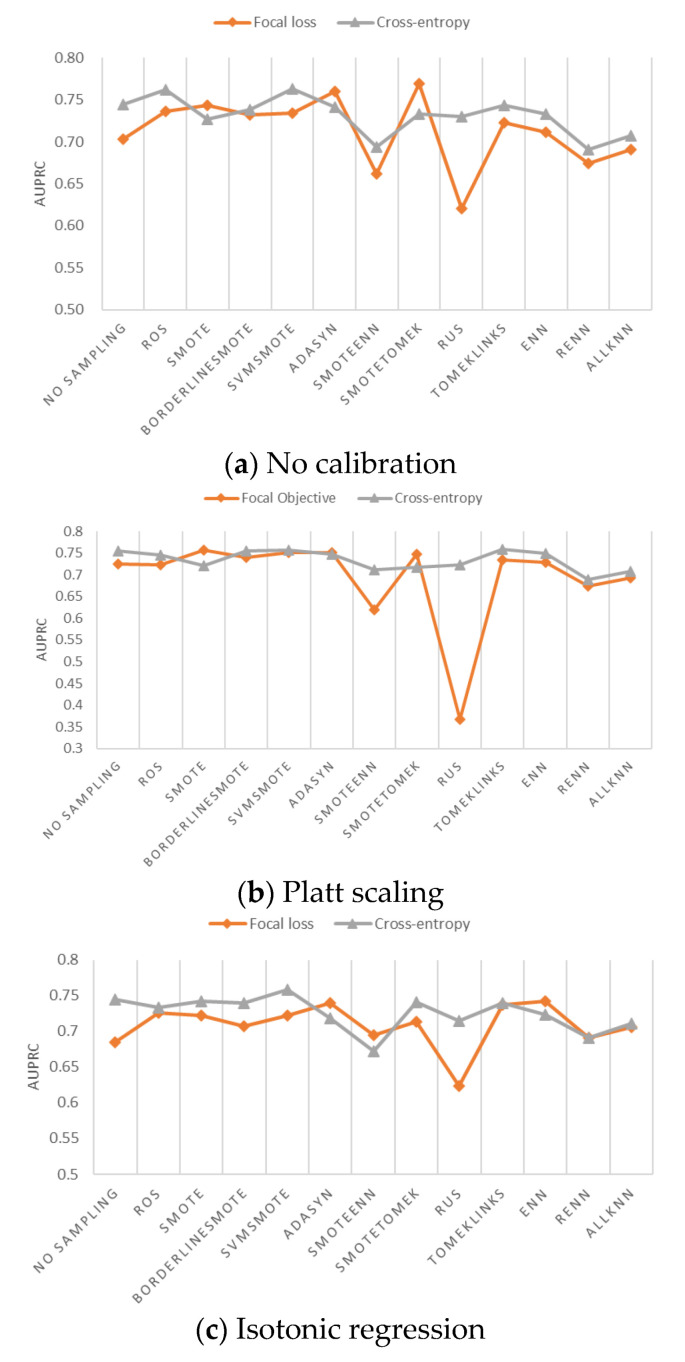
AUPRC of XGBoost with weighted focal loss and weighted cross-entropy loss.

**Figure 8 ijerph-18-07534-f008:**
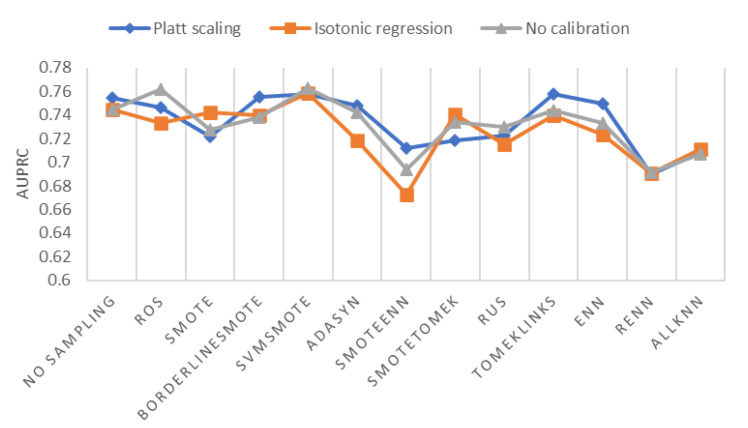
AUPRC of different sampling methods and probability calibration methods.

**Figure 9 ijerph-18-07534-f009:**
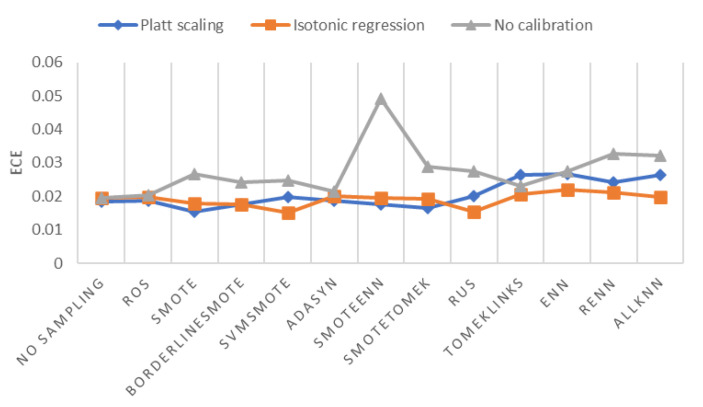
ECE of different sampling methods and probability calibration methods.

**Figure 10 ijerph-18-07534-f010:**
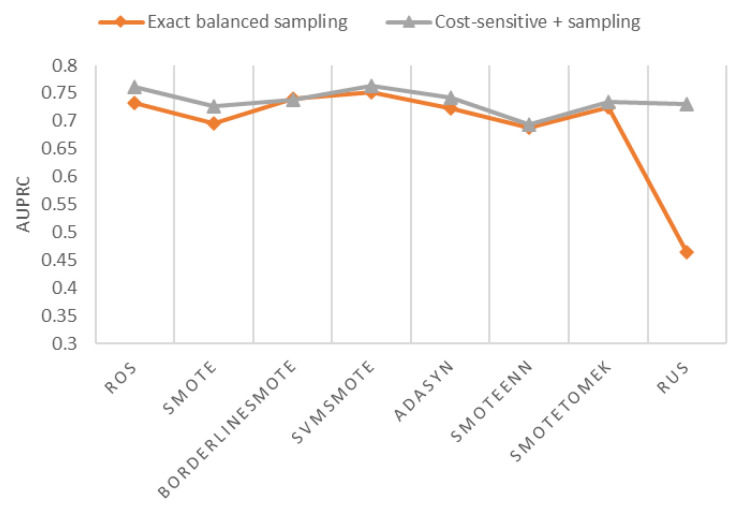
AUPRC of exact-balance sampling and sampling + cost-sensitive.

**Table 1 ijerph-18-07534-t001:** Hyperparameters to optimize.

No.	Hyperparameter	Definition	Parameter Range
1	n_estimators	Number of boosting rounds	(10,350), must be an integer
2	max_depth	Maximum tree depth for base learners	(3,10), must be an integer
3	learning_rate	Boosting learning rate	(0.1, 1)
4	subsample	Subsample ratio of the training instance	(0.5, 1)
5	colsample_bytree	Subsample ratio of columns when constructing each tree	(0.5, 1)
6	focal_gamma ^1^	Focal loss focusing parameter	(0, 4)

^1^ focal_gamma is only applicable when XGBoost uses focal loss as its loss function.

**Table 2 ijerph-18-07534-t002:** Five different sets of AutoML results.

Round	1	2	3	4	5
AUPRC (test data in CV1)	0.794	0.802	0.805	0.797	0.799
AUPRC (test data in CV2)	0.76	0.747	0.774	0.758	0.754
Sampling method	SVMSMOTE	SVMSMOTE	SVMSMOTE	SVMSMOTE	SVMSMOTE
Sampling ratio	0.26	0.49	0.36	0.34	0.57
Loss function	Cross-entropy loss	Cross-entropy loss	Cross-entropy loss	Cross-entropy loss	Cross-entropy loss
Minority example weight	14.89	8.72	22.01	4.08	16.02
Probability calibration	Platt scaling	No calibration	No calibration	Platt scaling	Isotonic regression
Focal_gamma	-	-	-	-	-
n_estimators	135	163	323	172	226
max_depth	5	7	6	7	8
learning_rate	0.33	0.52	0.13	0.40	0.10
subsample	0.62	0.81	0.53	0.84	0.58
colsample_bytree	0.63	0.75	0.51	0.58	0.68

**Table 3 ijerph-18-07534-t003:** Results of different sampling methods with weighted cross-entropy loss function and isotonic regression.

Sampling Method	AUPRC	Precision	Recall	F1	*ECE*
None	0.745	0.785	0.587	0.654	0.019
ROS	0.733	0.787	0.570	0.648	0.020
SMOTE	0.743	0.773	0.585	0.649	0.018
Borderline-SMOTE	0.740	0.763	0.562	0.625	0.017
SVMSMOTE	0.758	0.797	0.536	0.622	0.015
ADASYN	0.718	0.811	0.524	0.623	0.020
SMOTE-ENN	0.673	0.675	0.561	0.597	0.020
SMOTE-Tomek	0.741	0.745	0.608	0.652	0.019
RUS	0.715	0.784	0.527	0.619	0.016
Tomek Links	0.740	0.764	0.557	0.622	0.021
ENN	0.723	0.734	0.586	0.630	0.022
RENN	0.691	0.701	0.532	0.591	0.021
AllKNN	0.711	0.719	0.572	0.613	0.020

**Table 4 ijerph-18-07534-t004:** Results of different ACR threshold values.

ACR Threshold	Percentage of Risky Drivers	AUPRC	Precision	Recall	F1
0.1	30.05%	0.934	0.838	0.893	0.864
0.2	17.77%	0.910	0.825	0.791	0.801
0.3	11.09%	0.807	0.756	0.743	0.741
0.4	6.92%	0.822	0.756	0.678	0.713
0.5	4.29%	0.758	0.797	0.536	0.622
0.6	1.53%	0.553	0.628	0.400	0.447

## Data Availability

The vehicle trajectory data used in this study is free for non-commercial use and can be requested to download at https://www.highd-dataset.com (accessed on 1 May 2021).

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
