# Peer review of "Risky Driver Recognition with Class Imbalance Data and Automated Machine Learning Framework"

_ijerph, 2021, doi:10.3390/ijerph18147534_

Round 1

Reviewer 1 Report

The paper proposes a framework to identify risky drivers based on certain maneuvers, using trajectory data. It is well written, and easy to follow. Following are a few questions/concerns/suggestions:  - I suggest using "under-sampling" instead of "pre-sampling" (looks like in the methodology section you used under-sampling) - There are minor writing issues; please do a proof-reading before the next submission. Overall the paper is well written. - The paper says: "vehicles. Drivers who have relatively higher ACR than others in the same traffic environment are labeled as risky drivers". How would you define "relatively higher"? - Instead of using DFT to convert trajectory data to coefficients, wouldn't it also make sense to segment trajectories, build representations for fixed-size segments, and feed them to your model? To this reviewer, segmentation is a more common solution for trajectory data (that could have any length). - It can add more clarity if you can provide numerical examples of raw and processed input data. - Based on your description in section 2.7, you first find the optimal parameters based on 5-fold (CV1), and then run the final evaluation based on the best hyper-parameters found in the previous step using another 5-fold (CV2). But, this sort of model fitting and then evaluation is a bit problematic, since you might use the same data to find the best parameters, and then evaluate the model. A better way of doing this is to keep a part of your data as a held-out set, do your model/parameter fitting on the rest of data, and then do your final eval on the held-out. Please clarify if my understanding of your approach is incorrect, or if you believe otherwise (i.e. there is no chance of overfitting). - It would be nice to know how accurate is the quality of derived features (i.e. speed, acceleration, lane-changing, and car following) from the trajectories. - Is the entire trajectory data just for ~ 20 mins? - How did you come up with the threshold of 0.5 for ACR to label risky drivers? And how would your analysis/results change if we were to use other threshold values? A quick analysis would add additional values to this end. - My understanding is that you report the metrics on the positive (minority) class only, right? If that's the case, why don't you also report your framework's results on the negative (majority) class? - It would be nice to add a small section in which you can describe what other problems can be solved by your proposed framework.

Reviewer 2 Report

General comments

The paper deals with an extremely interesting subject, providing an autoML framework for risky driving classification using an ensemble method with XGBoost at its core. It uses an appropriate methodological approach and produces promising and novel research results. In addition, the paper is very well written; the authors are commended for their good effort. There are minor pending issues in the present form of the paper that should be addressed before it is accepted for publication.

  • In lines 48-55, several data recording methods are recorded. However, a major trend is smartphone data collection, which should be mentioned in the present context by the authors. See for instance:
  1. Kirushanth, S., & Kabaso, B. (2018). Telematics and road safety. In 2018 2nd International Conference on Telematics and Future Generation Networks (TAFGEN) (pp. 103-108). IEEE.
  2. Ziakopoulos, A., Tselentis, D., Kontaxi, A., & Yannis, G. (2020). A critical overview of driver recording tools. Journal of safety research, 72, 203-212.
  • On Figure 1, stratified 5-cross validation is ‘hard coded’. Would there be a possibility to include the number of folds in the optimization process and not have them fixed at 5?
  • On line 88, change ‘crashes’ to ‘accidents’ since that is the preferred term of this manuscript.
  • On lines 127, and on Figure 2, and on the following instances, the authors probably mean ‘calculate the rear-end collision risk (or potential or a similar description)’. If that is not the case please elaborate.
  • On lines 142-143, how were τ values selected? 0.7 s is the optimal human limit, so perhaps that is a very optimistic estimation.
  • On Table 1, the reviewer was surprised not to find alpha, lambda, max_delta_step and min_child_weight amongst the hyperparameters considered for tuning. Please explain that choice.
  • In the conclusions, Deep NN could be mentioned as well. Was XGBoost preferred due to its flexibility and lightness?

Reviewer 3 Report

The purpose of this paper is to propose a novel automated machine learning framework that simultaneously and automatically searches for the optimal sampling, cost-sensitive loss function, and probability calibration to handle class-imbalance problem in recognition of risky drivers. The topic is innovative and the methodology is also suitable for the research. Taking into account this positive assessment as a whole of the paper, I would like to mention some minor changes (or reflections) that the paper requires in order to be published. While methodology and results are quite well presented, the state of art needs to be improved

  • Abstract

The abstract is long and a little bit confusing. Please, avoid abbreviations in the abstract (like XGBoost or SVMSMOTE). If used in the rest of the text, please explain the meaning in brackets. Data base should be defined clearly as case study (data used in this paper was recorded on a 6-lane highway at the morning traffic peak. The trajectory of 2850 vehicles was recorded over 19 minutes 38 seconds9

  • Introduction

Page 3. At the beginning of the page, authors say: …. “UAV video-extracted vehicle trajectory dataset is used to train the model”. First explain the abbreviation UAV. Secondly, please clarify the country where the data set has been collected (vehicle trajectory- Germany).

  • Methodology

Page 4. In the research, authors consider only rear-end collision risk since “it is the most common collision risk” (according to Tak et al. 2018).  Do authors refer to the most collision risk in interurban roads? What about urban roads (streets)?

  • Data

The section of data should be shown before methodology in order to better understand the methodology.  Traffic was recorded at six German highways using unmanned aerial vehicles (UAV). Only private cars trajectory was recorded. What about motorcycles? Why this type of vehicles was nor studied? Why only the morning peak hour?
